# Determination of the Respiratory Compensation Point by Detecting Changes in Intercostal Muscles Oxygenation by Using Near-Infrared Spectroscopy

**DOI:** 10.3390/life12030444

**Published:** 2022-03-17

**Authors:** Felipe Contreras-Briceño, Maximiliano Espinosa-Ramirez, Vicente Keim-Bagnara, Matías Carreño-Román, Rafael Rodríguez-Villagra, Fernanda Villegas-Belmar, Ginés Viscor, Luigi Gabrielli, Marcelo E. Andía, Oscar F. Araneda, Daniel E. Hurtado

**Affiliations:** 1Laboratory of Exercise Physiology, Department of Health Science, Facultad de Medicina, Pontificia Universidad Católica de Chile, Santiago 7820436, Chile; maespinosa@uc.cl (M.E.-R.); vkeim@uc.cl (V.K.-B.); matias.carreno@uc.cl (M.C.-R.); rafael.rodrguez@uc.cl (R.R.-V.); fjvillegas@uc.cl (F.V.-B.); lgabriel@uc.cl (L.G.); 2Physiology Section, Department of Cell Biology, Physiology and Immunology, Faculty of Biology, Universitat de Barcelona, 08028 Barcelona, Spain; gviscor@ub.edu; 3Advanced Center for Chronic Diseases (ACCDiS), Division of Cardiovascular Diseases, Facultad de Medicina, Pontificia Universidad Católica de Chile, Marcoleta #367, Santiago 8380000, Chile; 4Biomedical Imaging Center, School of Medicine, Pontificia Universidad Católica de Chile, Santiago 7820436, Chile; meandia@uc.cl; 5Laboratory of Integrative Physiology of Biomechanics and Physiology of Effort (LIBFE), Kinesiology School, Faculty of Medicine, Universidad de los Andes, Santiago 7620001, Chile; ofaraneda@miuandes.cl; 6Department of Structural and Geotechnical Engineering, School of Engineering, Pontificia Universidad Católica de Chile, Santiago 7820436, Chile; dhurtado@ing.puc.cl; 7Schools of Engineering, Medicine and Biological Sciences, Institute for Biological and Medical Engineering, Pontificia Universidad Católica de Chile, Santiago 7820436, Chile

**Keywords:** exercise, near-infrared spectroscopy, respiratory muscles, oxygen uptake, respiratory compensation point

## Abstract

This study aimed to evaluate if the changes in oxygen saturation levels at intercostal muscles (SmO_2_-*m.intercostales*) assessed by near-infrared spectroscopy (NIRS) using a wearable device could determine the respiratory compensation point (RCP) during exercise. Fifteen healthy competitive triathletes (eight males; 29 ± 6 years; height 167.6 ± 25.6 cm; weight 69.2 ± 9.4 kg; $\dot{V}$O_2_-máx 58.4 ± 8.1 mL·kg^−1^·min^−1^) were evaluated in a cycle ergometer during the maximal oxygen-uptake test ($\dot{V}$O_2_-máx), while lung ventilation ($\dot{V}$E), power output (watts, W) and SmO_2_-*m.intercostales* were measured. RCP was determined by visual method (RCP_visual_: changes at ventilatory equivalents ($\dot{V}$E·$\dot{V}$CO_2_^−1^, $\dot{V}$E·$\dot{V}$O_2_^−1^) and end-tidal respiratory pressure (PetO_2_, PetCO_2_) and NIRS method (RCP_NIRS_: breakpoint of fall in SmO_2_-*m.intercostales*). During exercise, SmO_2_-*m.intercostales* decreased continuously showing a higher decrease when $\dot{V}$E increased abruptly. A good agreement between methods used to determine RCP was found (visual vs NIRS) at %$\dot{V}$O_2_-máx, $\dot{V}$O_2_, $\dot{V}$E, and W (Bland-Altman test). Correlations were found to each parameters analyzed (r = 0.854; r = 0.865; r = 0.981; and r = 0,968; respectively. *p* < 0.001 in all variables, Pearson test), with no differences (*p* < 0.001 in all variables, Student’s *t*-test) between methods used (RCP_visual_ and RCP_NIRS_). We concluded that changes at SmO_2_-*m.intercostales* measured by NIRS could adequately determine RCP in triathletes.

## 1. Introduction

To objectively estimate the exercise intensity in a non-invasive and straightforward way both during training and in a competitive race is crucial in long-distance sports, such as marathon, road cycling, or triathlon [1,2,3]. Coaches and athletes have traditionally used few physiological parameters, such as maximal heart rate (HR-máx.), oxygen uptake ($\dot{V}$O_2_-máx.), work rate (power (watts, W) or velocity), and lactate levels in capillary blood ([Lac]_blood_) [4,5]. Different models have been used to define training intensities zones according to changes in these parameters at systemic levels [6,7,8,9,10]. The gold-standard method to identify the training zones is based on the analysis of expired gases and changes at ventilatory parameters, determining ventilatory thresholds (VT1 or aerobic threshold, and VT2 or respiratory compensation point (RCP)) which corresponds to exercise-intensity where a particular source of energy supply has more prevalence [11]. However, it is expensive and difficult for athletes to access this kind of evaluation on a routine basis due to the high cost of the equipment and the high technical skills and training required of the staff. This gap must be covered by new methods that allow athletes to recognize the steady-state exercise intensity that they could maintain for a long time without limiting the blood flow and local oxygen delivery at peripheral locomotor muscles.

The assessment of oxygen saturation at muscular level (SmO_2_) has aroused growing interest to researchers in exercise physiology by the easy and not-expensive cost of the wearable devices that, by using the near-infrared spectroscopy (NIRS) principle, record the local changes of this parameter. Regarding, SmO_2_ has mainly been evaluated in locomotor muscles [12], showing good values for validity [13] and reliability [14,15]. Fontana et al. [16] found that the breakpoint at SmO_2_-*m.vastus laterallis* was associated with RCP in healthy participants during cycling exercise; and recently, Klusiewics et al. [17] determined that changes in SmO_2_-*m.vastus laterallis* are similar to maximal lactate steady-state in rowers. This evidence gives to SmO_2_ in locomotor muscles relevance and confidence to other traditional methods for prescribing exercise; however, this parameter can be conditioned by the energy demand of the respiratory muscles associated with the cost of breathing (COB) during exercise [18], a factor that varies among sports disciplines [19], and breathing patterns adopted during physical effort [20,21]. Considering that determination of VT is based on changes at ventilatory variables (mainly, lung ventilation ($\dot{V}$E)), it is relevant to know if changes at SmO_2_ of respiratory muscles (SmO_2_-RM) during exercise could help to determine training zones, and specifically the RCP. Our research group previously had been reported the good-to-excellent reliability of SmO_2_-RM measurements in long-distance runners [22], the impact associated with breathing patterns [21], and sex differences during exercise [23].

To our knowledge, few studies have been evaluated if the changes at SmO_2_-RM could identify ventilatory thresholds. Moalla et al. [24] reported that changes at SmO_2_-RM could identify RCP in children, similar to Rodrigo-Carranza et al. [25] in adult runners; however, these studies did not consider the possible influence of respiratory mechanics and breathing pattern adopted during physical effort on changes of SmO_2_-RM given the heterogeneous of their participants and type of exercise protocol used, as while as the determination of RCP was based on the v-slope method, which shows some difficulty to adequately identify RCP when there is no precise breakpoint from linearity in the data curve. To elucidate if oxygen level changes at respiratory muscles could determine RCP, this study aimed to evaluate the SmO_2_-*m.intercostales* using a wearable device (MOXY^®^) in fifteen competitive triathletes during a maximal incremental protocol exercise in a cycle ergometer, at the same time that exhaled gases and ventilatory parameters were recorded by the breath-by-breath method. Thus, parameters associated with exercising intensity, %$\dot{V}$O_2_-máx., $\dot{V}$O_2_, $\dot{V}$E, and power output were analyzed at exercise-time where RCP were determined by gold standard method and the breakpoint in SmO_2_-*m.intercostales* (RCP_NIRS_). We hypothesized that measuring changes of SmO_2_-*m.intercostales* during exercise is an adequate method to determine the respiratory compensation point and a valuable tool to prescribe training zones in athletes.

## 2. Materials and Methods

### 2.1. Design of Study and Participants

A cross-sectional observational study that assessed 15 healthy competitive triathletes (8 males; age 29.2 ± 5.5 years; height 167.6 ± 25.6 cm; weight 69.2 ± 9.4 kg; body mass index (BMI) 22.6 ± 1.8; maximum oxygen-uptake ($\dot{V}$O_2_-máx.) 58.4 ± 8.1 mL·kg^−1^·min^−1^; maximal cycling-load (watts, W) 318.8 ± 41.0 W) without a history of systemic problems (e.g., respiratory, cardiovascular, metabolic, musculoskeletal or neoplastic diseases) or any infectious or inflammatory process, for at least two weeks before measurements. The participants did not consume drugs, antioxidants, or any nutritional support. The sample size calculation was done by the software G*Power^®^ 3.1 (Heinrich-Heine-University, Dusseldorf, Germany) using previous data concerning the changes in oxygen saturation levels at *m.intercostales* from rest to respiratory compensation point (or anaerobic threshold, VT2) found in healthy subjects during cycling (from 77.5 to 64.0%) [23] and competitive marathoners during running (from 74.6 ± 10.7 to 52.9 ± 11.4%; and effect size 1.387) [21], considering a significance level of 5%, statistical power of 80% and two-tail test, plus considering 5% of data losing.

All participants were informed of the purpose, protocol, and procedures before informed consent was obtained from then. This study was approved by the Ethics Committee of the Pontificia Universidad Católica de Chile (Institutional Review Board, protocol number 210525001, date of approval: 9 September 2021). The study was carried out according to the Declaration of Helsinki for human experimentation.

### 2.2. Procedures

The measurements were done in the Laboratory of Exercise Physiology of the Pontificia Universidad Católica de Chile, under controlled environmental conditions (temperature 20 ± 2 °C; relative humidity 40 ± 2%) and fixed schedule (08:00 to 12:00 h). All participants were instructed to avoid physical activity 24 h before the day of evaluations and not to engage alcohol, caffeine or other stimulants and food intake for at least three hours prior to the measurements.

### 2.3. Anthropometric and Respiratory Measurements

The anthropometric assessments (body weight and height) were measured immediately after the participants arrived at the laboratory; subsequently, spirometry (Microlab, model ML3500, CareFusion^®^, San Diego, CA, USA) was performed according to the American Thoracic Society (ATS) and European Respiratory Society (ERS) protocols [26], utilizing the reference values of Knudson et al. [27].

### 2.4. Oxygen-Uptake Test

The maximal aerobic capacity ($\dot{V}$O_2_-máx.) was assessed by analyzing the ventilatory parameters (lung ventilation ($\dot{V}$E), respiratory rate (RR), tidal volume (Vt)) and exhaled gases (oxygen-uptake or consumption ($\dot{V}$O_2_) and carbon dioxide production ($\dot{V}$CO_2_)) by the *breath-by-breath* method (MasterScreen CPX, Jaeger^TM^, Traunstein, Germany) and expressed under standard temperature pressure dry air (STPD), while the participants completed an incremental exercise until voluntary exhaustion, despite verbal stimuli (respiratory quotient 1.20 ± 0.05). This test was performed on a bike connected to an electronically braked cycle ergometer indoor trainer device (KICKR^®^, Wahoo Fitness, Atlanta, GA, USA). The participants maintained the same position on their road bike throughout the entire exercise protocol (cycling postural position individually adjusted according to precedent training sessions). The exercise protocol consisted of a 2-min rest, 3-min warm-up period at 100 watts (W), followed by an increase of 20 W every 80 s until exhaustion or all criteria for ending the test were met. Participants were requested to maintain a cadence between 80 and 100 rpm during the test. To keep a good signal from the devices used (NIRS and ergoespirometer), participants were required to keep their arms on the handlebars during the test and not to adopt an “uphill” standing position of exclusive support on the pedals (without contact on the saddle) during the last stages.

The $\dot{V}$O_2_-máx. was calculated as the highest value obtained during the last 30 s of the incremental test, despite increasing the exercise intensity (<150 mL·min^−1^ of exercise) [28]. A cool-down phase of 3 min of the submaximal exercise was performed before the test. The heart rate, pulse oxygen saturation, blood pressure, and subjective sensation of physical effort by modified Borg´s scale were measured at baseline and throughout the test. Before every test, the gas analyzer was calibrated according to the instructions provided by the manufacturer.

### 2.5. Measurement of Muscle Oxygenation (SmO_2_)

During the protocol, oxygen saturation levels at *m.intercostales* (SmO_2_-*m.intercostales*) were non-invasively assessed using the monitor MOXY^®^ (Fortiori Design LLC, Hutchinson, MN, USA), which emits light waves close to the infrared range (Near-Infrared Spectroscopy, NIRS (630–850 nm)) from diodes to the surrounding tissue and records total haemoglobin (THb) and myoglobin (mHb) concentrations at the microvascular level [12]. This device detects the amount of emitted light that returns to two detectors placed 12.5 and 25.0 mm from the source, thus locally recording SmO_2_ through the interpretation of THb and mHb levels [29]. The light penetration depth is half of the distance between the emitting source and the detector (±12.5 mm) [30,31]. To determine SmO_2_-*m.intercostales*, a MOXY^®^ device was located in the seventh intercostal space of the anterior axillary line of the right hemithorax [21,22,23]. The device was fixed to the skin using the material suggested by the manufacturer, in addition to extra fixation with a cohesive band on the measurement zone, avoiding excessive compression that could alter the SmO_2_ record (see Figure 1). 

### 2.6. Data Analysis

Each athlete had an initial record of 90 s on the bike, followed by 60 s of baseline resting phase, during which data acquisition was synchronized. Using the ventilatory variables and exhaled gases values recorded by the ergoespirometer during exercise protocol, two blinded researchers determined by the visual method the respiratory compensation point (RCP_visual_) [11], which correspond to the secondary increase in $\dot{V}$E, the ventilatory equivalent of oxygen-uptake ($\dot{V}$E·$\dot{V}$O_2_^−1^), a marked increase in the ventilatory equivalent of carbon dioxide production ($\dot{V}$E·$\dot{V}$CO_2_^−1^) and decrease in the pressure value at the end of CO_2_ expiration (Pet-CO_2_) during exercise, above aerobic threshold (VT1). In case of a discrepancy between these evaluators, the opinion of an experienced third blinded research was possible, accepting as the definitive criterion that point at which at least two evaluators agreed [32]. None of the cases revealed differences between the two researchers in this study. Likewise, RCP_NIRS_ was determined using a segmented linear regression model on the SmO_2_-*m.intercostales* values recorded by MOXY^®^ [33]. Two linear segments were considered for the regression model, for which the segment slopes and breakpoint location were unknown variables. To estimate these variables, the sum of the squared differences of the segmented regression model with the data obtained in each subject is minimized. This provides numerical values for the time associated with the regression breakpoint for each subject, which is considered the threshold at which SmO_2_-*m.intercostales* decrease significantly. To determine if changes at SmO_2_-*m.intercostales* during exercise could identify the RCP, the values of %$\dot{V}$O_2_-máx., $\dot{V}$O_2_, $\dot{V}$E, and W obtained at RCP_visual_ and RCP_NIRS_ were analyzed. The values studied were the average of the last 30 s before RCP time (visual and NIRS). In this study, the data of SmO_2_ were expressed as a percentage ranging from 0 to 100%, considering the value at the resting stage as the maximal %SmO_2_-*m.intercostales* [31].

### 2.7. Statistical Analysis 

The normality of the data was evaluated using the Shapiro-Wilk test. To assess whether the variables analyzed differed between methods used to identify RCP (visual vs. NIRS), confidence intervals (95% CI) were calculated to determine the averages of the means of the differences of all the variables evaluated. The interpretation considered no differences when the null value was within the confidence interval (Bland-Altman test). Pearson’s correlation coefficient assessed correlations between variables’ values analyzed at RCP time (visual and NIRS). Also, a comparison of variables was performed using the paired Student´s *t*-test. Statistical significance was set at *p* < 0.05. The statistical analysis was performed using GraphPad Prism (version 8.0; San Diego, CA, USA). 

## 3. Results

The participants’ characteristics, spirometry parameters, and maximum cardiorespiratory values at the oxygen-uptake test are shown in Table 1. The lung function was normal with no restrictive or obstructive alterations. The oxygen-uptake test ended when all participants achieved criteria for maximal effort.

During exercise protocol, the ventilatory equivalent of carbon dioxide ($\dot{V}$E·$\dot{V}$CO_2_^−1^) showed an abrupt increase because of increased lung ventilation ($\dot{V}$E) associated with increased effort intensity. Concomitantly, the muscle oxygen saturation level at *m.intercostales* decreases markedly, corresponding with a breakpoint detected by the algorithm used (RCP_NIRS_). The THb levels measured by MOXY^®^ did not change during exercise protocol. Figure 2 shows the changes of variables analyzed during exercise in a participant.

### 3.1. Comparison between RCP_visual_ and RCP_NIRS_

No differences between variables selected to compare RCP_visual_ and RCP_NIRS_ methods were found (paired Student’s *t*-tests, see Figure 3).

### 3.2. Correlations

The Pearson´s correlations analysis for each variable studied is shown in Figure 4. A good-to-excellent direct association was found between both methods (*p* < 0.001, for all variables).

### 3.3. Agreement

The Bland-Altman plots are shown in Figure 5. The mean differences and limits of agreements against the average value obtained by both methods used to detect RCP (visual and NIRS) for %$\dot{V}$O_2_-máx., $\dot{V}$O_2_ relative (mL·kg^−1^·min^−1^), $\dot{V}$E, and load (watts, W) showed that the differences in scores were within 95% of the confidence interval, showing that both methods predict RCP closely.

## 4. Discussion

The study aimed to evaluate if changes in oxygen saturation levels at *m.intercostales* measured by a wearable device could accurately determine the respiratory compensation point (RCP) in competitive triathletes during cycling exercise. The main findings are that in variables associated with exercise-intensity (%$\dot{V}$O_2_-máx., $\dot{V}$O_2_ relative (mL·kg^−1^·min^−1^), $\dot{V}$E, and load (watts, W), were not significant differences between the traditional method used to determine RCP (visual method by analysis of exhaled gases and ventilatory parameters) and the breakpoint in %SmO_2_-*m.intercostales* during cycle exercise; good-to-excellent levels of associations; and good levels of agreement (high reliability). To our knowledge, this is the first study that reports that RCP could be determined by a wearable device (MOXY^®^) during cycling exercise in competitive triathletes.

Few studies have focused on identifying RCP using a NIRS device on respiratory muscles during exercise. Although, Moalla et al., (2005) in children [24], Fontana et al. (2015) [16] in healthy adults, and Rodrigo-Carranza et al., (2021) in runners [25] have reported interesting results, those data are difficult to extrapolate to our participants owing to factors such as the method used for RCP determination [11,32], sex-differences associated to ventilatory response to exercise [23], high biological variability of cardio-ventilatory responses during exercise in a non-steady-state [19], breathing patterns adopted during effort [21], intensity of physical exercise performed [20,34,35,36], decreased muscle perfusion as a consequence of muscular contractions [37,38], loss of SmO_2_ signal due to changes in positions and/or adipose tissue where the devices are positioned [39,40], and type of device used to record muscle oxygen levels [12,30,41,42]. In this study, we used a MOXY^®^ for recording SmO_2_-*m.intercostales.* This device has shown good values of validity [14] and reliability [22] to record SmO_2_ during exercise; and good agreement levels to other NIRS devices [30]. The depth of light penetration is maintained at near 12.5 mm, aspects that allowed the non-presence of artefacts in the assessment of *m.intercostales* in the triathletes of this study, given they were within the normal BMI range; this strength from the wearable device has been found previously by our research group in runners [21] and healthy subjects [23]. Another relevant factor is the method used to identify RCP. This visual method is based on the criterion used by experienced researchers, who determine RCP considering the abrupt increase in the ventilatory equivalent of carbon dioxide production ($\dot{V}$E·$\dot{V}$CO_2_^−1^) and decrease in the pressure value at the end of CO_2_ expiration (Pet-CO_2_) during exercise, above aerobic threshold [43]. Other studies with interesting findings have used other methods to identify RCP, such as the v-slope method, based on the breakpoint of the linearity of the curve $\dot{V}$CO_2_ vs. $\dot{V}$O_2_ [24,44,45], breakpoint at heart-rate variability [46,47,48,49,50], or lactate blood level curves [11,41,51]; however, in order to diminish the variability inter- and intra- evaluator, we chose the method widely used for exercise scientists, with good results because in no case was there a discrepancy between two experienced researchers.

Another finding was regarding the changes of SmO_2_-*m.intercostales* during the exercise protocol. This data showed abrupt decreases while $\dot{V}$E increased exponentially, sustaining that an increase of recruitment of these muscle fibers implicates a decrease in oxygen levels in respiratory muscles. The changes of oxygen levels in respiratory muscles during our exercise protocol were similar to previous studies [16,20,21,22,23,24,25,34], recognizing this assessment as a novel and non-invasive method to estimate the cost of breathing (COB) or oxygen-uptake associated to the work of the respiratory muscles during exercise in athletes. Concerning, the COB at rest is near 2% of the total oxygen requirement in healthy adults [52], while during exercise, by the increase of $\dot{V}$E, it reaches 9 ± 3% of $\dot{V}$O_2_-máx. in trained subjects [53,54], 10–15% in inactive subjects [55], and 30–45% in patients with chronic respiratory diseases [56,57,58]. At exercise-time where RCP_visual_ was determined, the %SmO_2_-*m.intercostales* decreased between 10–15%, reaching a greater decrease at maximal stage (30–35%); these results are similar to exercise-time at the breakpoint in which SmO_2_-*m.intercostales* was found by the linear regression model used, and maximal stage (data not showed).

In this study, both sex groups showed similar results (data not showed); however, possible sex differences is another aspect to include in future research because women have shown more COB at the same exercise intensity with lower values of SmO_2_ at respiratory muscles than their male counterparts [23,59,60,61]. The relatively low number of subjects and the homogeneity of participants’ characteristics in terms of their fitness level probably influenced the fact that no possible sex differences were found.

A limitation of this study is the lack of adipose thickness measures by skinfold and/or ultrasonography. This prevented us from verifying that the muscle tissue assessed was within the measuring range of the wearable device used to record SmO_2_ (MOXY^®^) [30]. In the case of the MOXY^®^ device, the light penetration is maintained at near 2 cm, aspects that allowed the non-presence of artefacts in the assessment of *m.intercostales* in the triathletes of this study, given they were within the normal BMI range. However, we consider that a limitation of the MOXY^®^ device is that it does not report the oxygenated and deoxygenated haemoglobin vs. myoglobin levels, and specially that is not possible to know the blood flow in the muscles assessed; this aspect should be evaluated in future studies to elucidate if the local SmO_2_ changes are caused by oxygen consumption or vasomotor responses. For instance, a decrease in SmO_2_ can be due both to local vasoconstriction and low delivery of blood flow or exclusively because of high oxygen uptake at muscle tissue, even under a higher blood flow regime. Another limitation is that the cyclic hormonal variations should be considered in women participants, given that oedema, dehydration, and altered thermoregulation are factors affecting physical performance by modifying the ventilatory center response [62,63,64]. New studies could consider the stage of the menstrual cycle; however, it is not consistently found to influence exercise performance [65,66]. Future research should include a larger cohort with a greater variety of long-distance competitors (e.g., runners, cyclists, ultra-trail, etc.) to confirm the results obtained in this study.

It is of interest that future studies will evaluate the impact of respiratory muscle training on the exercise-induced changes at SmO_2_-*m.intercostales*. To our knowledge, it is unknown whether specific training regimes, such as ones based on resistance like load threshold (e.g., POWERbreathe^®^, MT Technologies, Birmingham, UK) or endurance training like isocapnic hyperpnea (e.g., SpiroTiger^®^, MVM, Bologna, Italy), could have more beneficial effects either on this parameter or improvements on exercise capacity reflected in ventilatory threshold changes. Also, the associations between the assessments of respiratory muscles oxygen-uptake or oesophageal balloon catheter and SmO_2_-*m.intercostales* measurements can be explored to define the best and low-cost method that allows sports and clinicians to incorporate an easy and valuable way to measure the COB adequately.

## 5. Conclusions

This study demonstrates that assessing oxygen saturation levels at *m.intercostales* during exercise is a valid method to determine the respiratory compensation point (RCP) in triathletes. Our results show an adequate level of agreement, good-to-excellent levels of associations, and no differences between the gold-standard method to determine the RCP (the visual method by analysis ventilatory variables and exhaled breath gases), and changes of SmO_2_-*m.intercostales* recorded by a wearable device using NIRS principle. These findings are helpful for athletes and coaches, allowing to recognize adequately the exercise intensity or training zone at which to perform or prescribe sport activities, disregarding not-easy access and expensive techniques.

## Figures and Tables

**Figure 1 life-12-00444-f001:**
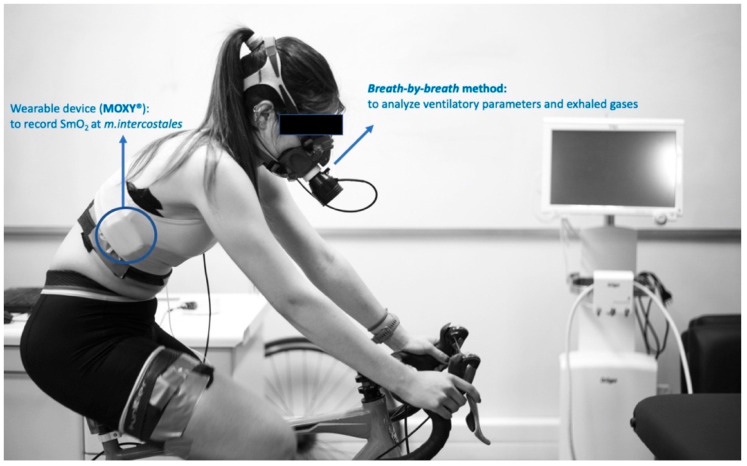
Scheme design.

**Figure 2 life-12-00444-f002:**
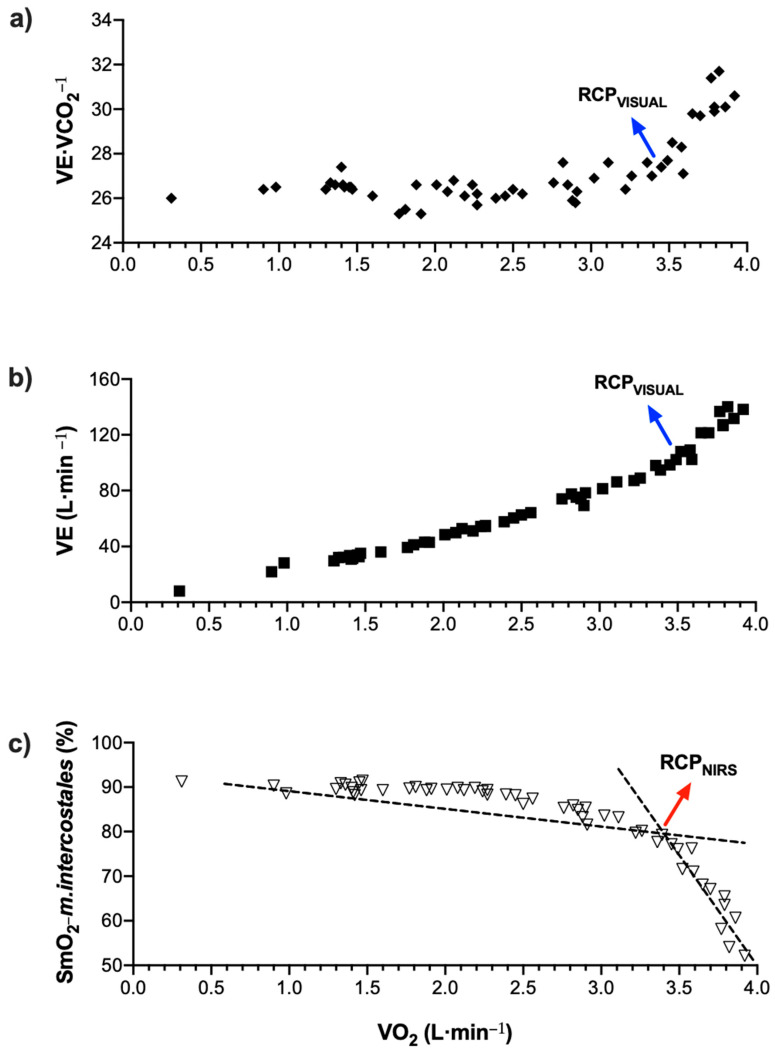
Example of RCP determination in a participant: (**a**) and (**b**) RCP_visual_ and (**c**) RCP_NIRS_.

**Figure 3 life-12-00444-f003:**
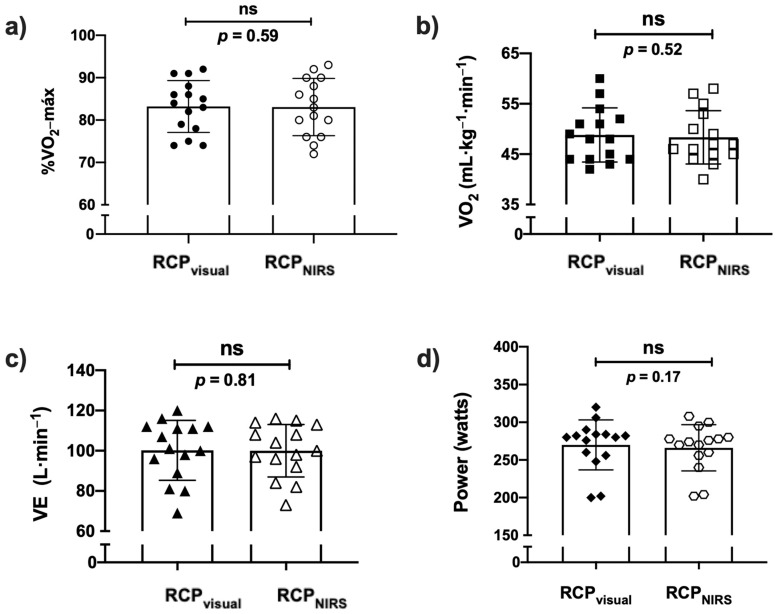
Comparison of each variable analyzed at RCP in both methods (visual and NIRS): (**a**) %$\dot{V}$O_2_-máx.: percentage of maximum oxygen-uptake; (**b**) $\dot{V}$O_2_ relative: oxygen-uptake expressed as a relative value (mL·kg^−1^·min^−1^); (**c**) $\dot{V}$E: lung ventilation (L); and (**d**) Watts: Load of ergometer.

**Figure 4 life-12-00444-f004:**
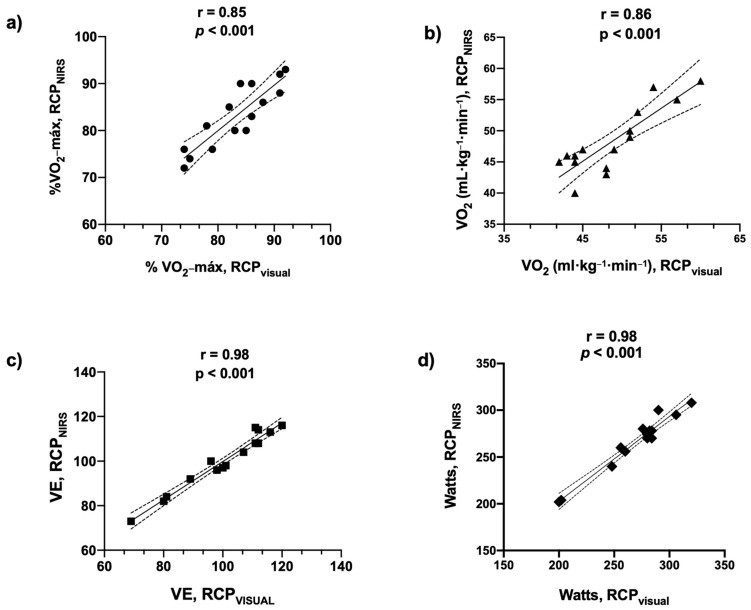
Associations of each variable analyzed at RCP in both methods (visual and NIRS): (**a**) %$\dot{V}$O_2_-máx.: percentage of maximum oxygen-uptake; (**b**) $\dot{V}$O_2_ relative: oxygen-uptake expressed as a relative value (mL·kg^−1^·min^−1^); (**c**) $\dot{V}$E: lung ventilation (L); and (**d**) Watts: Load of ergometer.

**Figure 5 life-12-00444-f005:**
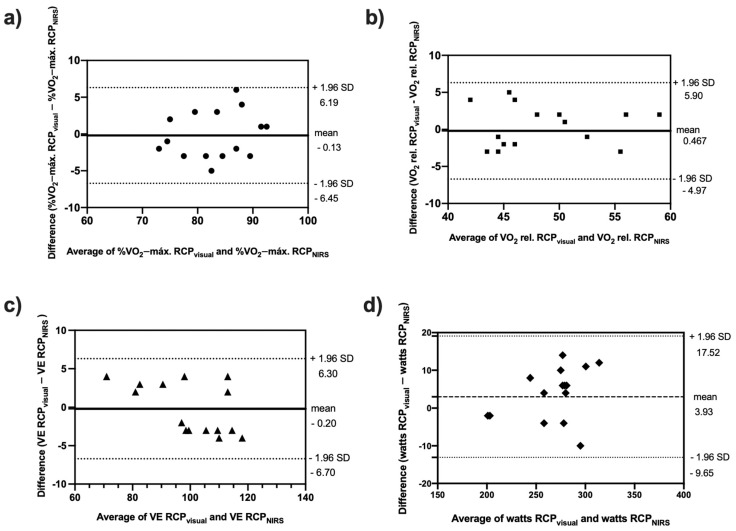
Plots of Bland-Altman tests of variables values at RCP in both methods (visual and NIRS): (**a**) %$\dot{V}$O_2_-máx.: percentage of maximum oxygen-uptake; (**b**) $\dot{V}$O_2_ relative: oxygen-uptake expressed as a relative value (mL·kg^−1^·min^−1^); (**c**) $\dot{V}$E: lung ventilation (L); and (**d**) Watts: Load of ergometer.

**Table 1 life-12-00444-t001:** Characteristics of participants and maximum cardiorespiratory values at oxygen-uptake test.

Variables	Mean ± Standard Deviation
sex (n)	male = 8; female = 7
years	29.2 ± 6.5
height (cm)	167.6 ± 25.6
weight (kg)	69.2 ± 9.4
BMI	22.6 ± 1.8
Triathlon experience (year)	8.2 ± 2.3
Training volume (hours·week^−1^)	18.0±2.3
FEV_1_ (L)	4.28 ± 0.78
FEV_1_ (% predicted)	94.0 ± 1.2
FVC (L)	5.03 ± 1.03
FVC (% predicted)	111.9 ± 2.5
FEV_1_ ·FVC^−1^ (%)	85.0 ± 7.5
load-máx. (watts)	318.8 ± 41.0
$\dot{V}$O_2_-máx. (mL·kg^−1^·min^−1^)	58.4 ± 8.1
$\dot{V}$E-máx. (L·min^−1^)	168.4 ± 29.3
HR-máx. (bpm)	184.2 ± 8.6
%HR-máx. (220-age)	96.0 ± 1.8
RPE	9.8 ± 0.4

**Abbreviations**: BMI = body mass index; FEV_1_ = Forced expiratory volume at first second; FVC = Forced vital capacity; $\dot{V}$O_2_-máx = maximum oxygen uptake; $\dot{V}$E = lung ventilation; HR = heart rate; RPE = Rate of perceived exertion (assessed by modified Borg scale).

## Data Availability

Not applicable.

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
