# Peer review of "Determination of the Respiratory Compensation Point by Detecting Changes in Intercostal Muscles Oxygenation by Using Near-Infrared Spectroscopy"

_life, 2022, doi:10.3390/life12030444_

Round 1

Reviewer 1 Report

In this study, the authors have determine oxygen saturation levels at m.intercostales during exercise using a wearable device. The study is mostly complete. The conclusions drawn are well supported by the result. The authors have also discussed the limitation of the study. The quality of presentation is good and the sample size is adequate. I support the publication of the manuscript.

Author Response

Thank you for your valuable opinion and kind comments.

Reviewer 2 Report

The study by Contreras-Briceño et al. compares two methods of determining the respiratory compensation point during cycle exercise. The comparison is made between near-infrared spectroscopy (NIRS) of the intercostal muscles via the MOXY device against visual inspection via expert analysis. The authors concluded the MOXY system can identify the respiratory compensation point similarly to visual inspection.

There are some concerns with the description of the methods used to collect the NIRS data. Clarifying these points would give readers confidence that the data were not influenced by confounding factors such as body composition and non-respiratory activity.

General comments

  1. The authors frame the results in terms of ‘oxygenation’ where the methods state the outcome of interest was total-haemoglobin (THb). Why was THb chosen as opposed to oxygenated Hb or tissue oxygenation/saturation index?

  1. Is it possible the NIRS signal can be counfounded by the viscera given the proximity between intercostal muscles and organs such as the liver? How have the authors controlled or accounted for this?

  1. The muscles over the intercostal space also serve postural function. What measures were taken to ensure posture was maintained throughout the exercise test? Any twisting

  1. Have the authors considered sex differences as it relates to this study? There are previously reported differences in how males and females use respiratory muscles during cycling. Could these differences have affected the results of the study?

  1. Figures: There is some inconsistency in the axes of the figures. In some cases the axis starts at 0, shows a break and then continues at the relevant point, whereas others do not begin at zero.

  1. The authors spend time in the discussion and introduction explaining their NIRS outcomes are cheap and effective for training purposes. How do the outcomes in the current study compare to ratings of perceived exertion?

  1. Despite noting the lack of adipose thickness measurements as a limitation to the current study, more detail is required to assure the reader that this did not affect the sensitivity of the measurement technique. BMI alone is not sufficient to say adipose tissue thickness would not affect the NIRS signal, as BMI does not describe the distribution of body fat. Moreover, body composition is different between males and females, and could also affect adipose thickness over areas of interest.

Specific comments:

Line 116: The authors are commended for including a sample size calculation in their methods. However; the data provided are not enough to replicate the calculation.

Line 145: Were participants allowed to bring their own bike or was the same lab bike used for all participants?

Line 152: Please clarify ‘upstairs position’.

Line 285: Unclear what ‘mode-o’ type of exercise. Please define.

Line 291: The authors state light penetration was constant at 2 cm. However earlier (Line 168) states light penetration is half the distance between emitter and detector. The greatest distance between emitter and detector noted was 2.5 cm. Therefore the light penetration would be 1.25 cm at maximum correct?

Line 308: Do the authors have a physiologic explanation as to why the oxygen delivery to muscles would decrease when presumably performing more work?

Line 310: The cited studies are from the same group as the present study. Are there instances of other groups reaching similar conclusions?

Line 316: The description of how smO2 data were normalized would better fit in the methods.

Line 325: This line is not clear. As read I understood it to say having a high-aerobic capacity did not affect the results of the study but the explanation offered is due to the high training status of the participants. It is not clear how the authors have determined that high fitness did not affect the NIRS and RCP results of the study.

Line 345: Assuming the sample size calculation presented earlier in the manuscript is accurate, I do not think it is worth highlighting the low sample size as a limitation to the present study. While I agree it would benefit future research to have a larger and more varied sample the authors have provided evidence that the sample for this study is adequate.

Table 1: Consider including spirometry values as a percent of predicted values as well as absolute values. Perhaps also for heart rate to as heart rate maximum could be influenced by age?

Figure 2: Adding lines of fit to panels (A) and (B) may help the reader visualize the identification of RCP similar to what was done for panel (C).

Figure 2: Units for panel (C) y-axis.

Round 2

Reviewer 2 Report

The authors have done well to respond to the original comments presented in text.

Please check the figures. Despite noting figures have been changed the version of the manuscript for revision appeared to have the original figures before any revision.

Author Response

Reviewer #2: round 2

The authors have done well to respond to the original comments presented in the text. Please check the figures. Despite noting figures have been changed the version of the manuscript for revision appeared to have the original figures before any revision.

Answer: Thank you again for this relevant comment. In the new version of this manuscript, we incorporated the figures changed after round 1 of this review process. We appreciate all your comments and suggestions that have contributed to reaching an improved version of the original manuscript.